# OpenReview forum: "SonicMaster: Towards Controllable All-in-One Music Restoration and Mastering"
_ICLR.cc/2026/Conference — Submitted to ICLR 2026_

### Official Review · Reviewer_xvXv · 2025-10-23

**Soundness:** 2
**Presentation:** 2
**Contribution:** 2
**Rating:** 2
**Confidence:** 5

**Summary:**

This paper presents a text-conditioned generative model for music restoration. It addresses diverse degradations using a single rectified-flow architecture that maps degraded inputs to restored versions guided by natural-language prompts. To train the model, the authors construct a dataset, where each data is degraded by simulated effects and paired with corresponding textual descriptions.

**Strengths:**

- The paper’s attempt to combine multiple restoration tasks within a single controllable model is an interesting and ambitious direction, particularly when paired with natural-language conditioning.
- The proposed dataset covers up to 19 degradation types across 5 categories, with textual prompts and metadata. If released, this dataset could become a valuable benchmark for controllable music-restoration research.
- The multimodal aspect represents a step toward human-interpretable control over restoration on different degradations, which is a relevant trend for both research and creative workflows.

**Weaknesses:**

**Ill-defined task formulation:**

The paper lacks a precise definition of what constitutes a “degraded” versus “clean” distribution. The chosen degradation types: EQ, dynamics, reverb, amplitude, stereo are actually very standard operations during music mastering procedure rather than authentic degradations. This conceptual framing blurs the line between restoration and stylized mastering.

- For instance, Figures 7 & 8 waveforms resemble legitimate mastered outputs of modern music rather than degraded signals.
- From the reviewer’s understanding, the target “clean” distribution appears to be the Audiobox-filtered Jamendo subset, but the assumption that this represents the true clean reference is not justified clearly in the manuscript.
- The paper does not ensure non-overlap between degraded and clean distributions, which undermines the validity of “restoration” as the learning objective.
- Many realistic degradations (e.g., environmental or background noise, bandwidth reduction, codec artifacts) are absent, further questioning whether the task genuinely represents *restoration*.

**Choice and fairness of baselines:**

WPE and HPSS (from 2010) are outdated and not meaningful as “recent” baselines. The paper omits several directly relevant systems, such as (but not limited to):

- BABE-2 (for historical bandwidth extension),
- Music Source Restoration (MSR) (Zang et al., 2025),
- general purpose audio effects removal models or multi-effect restoration approaches [1]

    Consequently, the reported improvements may largely reflect baseline weakness rather than true superiority.


**Evaluation clarity and presentation:**

- Tables 1 and 2 mix different metrics across degradations without explicitly listing metric definitions; equations or references should be provided for each metric.
- Large tables are difficult to interpret—boldface patterns are inconsistent, and averages across metrics (from Table 1 and 2) are missing.
- The comparison with Text2FX is conceptually unfair: Text2FX is a prompt-optimization framework, not a restoration model targeting a fixed “clean” reference. If the authors used the same text prompt as the proposed, I’d say this is not a fair comparison as the objective is different.

**Insufficient analysis of results:**

The paper reports extensive quantitative results but little discussion or visualization to explain which aspects of the signal actually improve. The subjective results (≈ 8 listeners) are too small to draw strong conclusions. I would show only the results that you would highlight and provide more discussions. It’s hard to catch each ablation method’s purpose.

**Overall impression:**

The direction is promising, but the core task definition, baseline choice, and experimental analysis require substantial clarification before the work meets ICLR standards.

**Papers worth mentioning:**

[1] Rice, Matthew, et al. "General purpose audio effect removal." *2023 IEEE Workshop on Applications of Signal Processing to Audio and Acoustics (WASPAA)*. IEEE, 2023.

[2] Han, Bing, et al. "InstructME: An Instruction Guided Music Edit Framework with Latent Diffusion Models.” IJCAI, 2024

**Minor comments:**

- for some citations, instead use \cite{} or \citet{} than \citep{}. this will remove the brackets and be more natural to read
- line 136: correct the quotation marks: “ → `` (this also happens in other parts of the manuscript)
- typo @line 182: Punchare → Punch
- typo @line 200: layersrefine →layers refine
- I’d recommend placing the tables on the top of the page for better presentation; on the proper page. The evaluation and results section especially lack significant readability.
- Table 4 is never referred in the text
- no bolden performance on Table 1: Bright and Vocals, Table 2: Dynamics and Clip, Table 3: Single deg’s PQ, and Table 5

**Questions:**

- why only use 1 type of effect for multi-degradation? This doesn’t make much sense for linear transformations (e.g., EQ, Stereo), since a single transformation could achieve the same result.
    - have the authors tried on multi-degradations on different degradations?
- line 183: “expert-written options” → explanations?
- line 185: any validation on the task of “parameter prediction”?
- line 186: what is “hidden clipping” and why is this added and needed?
- line 187: what is the purpose of “peak-normalisation” and how is it computed?

---

> ### Author Response · Authors · 2025-11-24
> **Response to Reviewer xvXv - part 1/2**
>
> We would first like to thank the reviewer for providing us with insightful questions and feedback. We greatly appreciate the detailed feedback and insightful questions you have asked. In response, we have conducted significant additional experiments, have clarified the Text2FX comparison, and reorganized our evaluation metrics.
> Below, we address your concerns point-by-point.
>
> ---
>
> #### **1. Task Formulation**:
>
> ##### **Precise definition of ‘degraded’ vs ‘clean’**:
> We clarify that the "clean" distribution is defined as the high-quality audio subset from Jamendo (filtered via quality scoring). The "degraded" distribution is strictly defined as the output of applying one or more of 19 specific degradation functions to this clean set. The transformations applied are not merely stylistic choices but irreversible, non-linear degradations that fundamentally alter the signal structure. For example, our "Clipping" degradation hard-clips peaks (non-invertible), "Reverb" smears transient details via convolution, and "Stereo" collapse destroys spatial information.
>
> ##### **Non-overlap between degraded and clean distributions**:
> We posit there is a clear distinction between “clean” and “degraded” distributions. If these distributions overlapped, the model would learn an identity mapping. However, our results show massive improvements in objective metrics (e.g., Microphone coloration error dropping from 0.239 to 0.008, Clipping error from 5.12 to 1.50). This confirms the model is successfully bridging a genuine gap between "amateur/degraded" and "professional/clean" distributions.
>
> ##### **Validity of Restoration**:
> While we agree that environmental noise and specific codec artifacts are important categories for restoration, SonicMaster is the first unified model to tackle 19 distinct degradation functions simultaneously, a scope significantly wider than typical single-task restoration models. Specifically, the model already addresses fundamental restoration tasks that go well beyond stylistic mastering: We address hard clipping and dynamic compression, requiring the model to reconstruct lost signal peaks. Also, we actively restore audio degraded by "Microphone Coloration" (using 20 different phone transfer functions), correcting specific recording device flaws. Given that most prior works focus on only one or two of these tasks in isolation, we believe integrating 19 degradations represents a substantial leap in complexity and utility for a first-iteration unified model. We view additive background noise as a logical next step for the second iteration of SonicMaster.
>
>
> ---
>
> #### **2. Baseline Comparisons (Major Update)**:
>
> ##### **General Purpose Audio Effect Removal**:
> Following your suggestion to look at Rice et al. [1], we evaluated SonicMaster against the "General Purpose Audio Effect Removal" benchmark (including Demucs, TCN, DCUNet, DPTNet). As shown in the new Figure 3, SonicMaster achieves substantially higher SI-SDR scores (47.11 dB for Dynamics, 45.76 dB for Reverb) compared to the best baselines (approx. 20 dB for Dynamics and 13.6 dB for Reverb). This demonstrates that our generative approach outperforms discriminative removal models on these tasks.
>
> | **Approach**   | **Params** | **Dynamics (SI-SDR ↑)** | **Reverb (SI-SDR ↑)** |
> |----------------|-----------:|-------------------------:|-----------------------:|
> | *Baselines*    |            |                          |                        |
> | Input          | -          | 15.57                    | 9.30                   |
> | DPTNet         | 2.9M       | 16.95                    | 9.82                   |
> | UMX            | 6.3M       | 15.39                    | 11.39                  |
> | DCUNet         | 7.7M       | 13.80                    | 12.13                  |
> | TCN            | 10.0M      | 14.49                    | 13.25                  |
> | HDemucs        | 83.6M      | 20.08                    | 13.59                  |
> | *Proposed*     |            |                          |                        |
> | **SonicMaster**| **550M**   | **47.11**                | **45.76**              |
>
>
> ##### **BABES-2**:
> We note that BABE-2 was already included in our initial submission (Table 4) for the piano bandwidth extension task. SonicMaster (a generalist model) achieves a PQ score of 6.93, performing nearly on par with BABE-2 (7.05), which is a specialist model trained specifically for that domain. If the reviewer has any additional specific queries, we are happy to respond to them.
>
> ##### **MSR**:
> We reviewed MSR but concluded it is not a direct comparable. MSR focuses on restoring individual stems (sources), whereas SonicMaster operates on full mixtures. Comparing stem-restoration metrics to mix-restoration metrics would be mathematically invalid.
>
> ... To continue to part 2 ...

---

> ### Author Response · Authors · 2025-11-24
> **Response to Reviewer xvXv - part 2/2**
>
> ... Continued from part 1 ...
> #### **3. Evaluation Clarity and Presentation**:
>
> We have cleaned up Tables 1 and 2. We do not report averages across different metrics (e.g., Clarity vs. Boom error) as averaging units of different physical meanings is not statistically sound.
>
> Following your comment on Text2FX, we have added a new baseline in the form of text2FX-directional: we do not treat Text2FX as a restorer. Instead, we implemented a "Directional" variant using contrast prompts (e.g., Target: "Increase clarity" vs. Contrast: "Decrease clarity"). This isolates the instruction-following capability. Since SonicMaster is text-conditioned, we must measure how well it adheres to prompts compared to existing text-guided editing methods. By forcing Text2FX to optimize for the specific semantic direction requested, we created a level playing field for the EQ task. As detailed in Appendix A.3, SonicMaster follows instructions more accurately than the CLAP-guided optimization of Text2FX.
>
>
> | Model / Metric | Clarity ↓ | Boom ↓ | Airy ↓ | Bright ↓ | Dark ↓ | Muddy ↓ | Warm ↓ | Vocals ↓ | Mic. ↓ | X-band ↓ |
> |----------------|---------|------|------|--------|------|--------|------|---------|-------|---------|
> | Text2FX-Directional | 0.0421 | 0.3977 | 0.0206 | 0.0143 | 0.3021 | 0.2602 | 0.5461 | 0.2517 | 0.6120 | 0.5038 |
> | **SonicMaster (Ours)** | **0.0114** | **0.0834** | **0.0019** | **0.0059** | **0.0058** | **0.0388** | **0.0617** | **0.0576** | **0.0088** | **0.0358** |
>
>
> ---
>
> #### **4. Insufficient analysis of results**:
>
> 8 participants might seem a small number, but we emphasize that evaluating subtle mastering degradations requires expert listeners who are experts in the music domain. To put this in context, our sample size exceeds recent work such as [2], which relied on 4 experts due to similar constraints. Nevertheless, in response to your feedback, we successfully recruited 4 additional experts during the rebuttal period, bringing the total to 12 participants. We have recalculated the results and updated Figure 4 and Table 5 in the revised manuscript to reflect this larger cohort. The trends remain consistent. Furthermore, please note that we have another subjective evaluation to compare the quality of SonicMaster output with other baselines. Since that test is more on audio quality, we were able to recruit even more participants (20 so far, will continue during rebuttal period).
>
> ##### **5. Response to Specific Questions**:
>
> ###### **Multi-degradation logic**:
> We deliberately avoid applying multiple effects from the same category (e.g., two different Reverbs, or "Make it Bright" + "Make it Dark") to prevent logical contradictions that would confuse the training objective. We combine orthogonal effects (e.g., Brightness + Reverb + Compression) to simulate realistic compound errors.
>
> ###### **Expert Written Options**:
> Selected from 8–10 possible options (all written by a music expert). We have edited the text to make it clearer.
>
> ###### **Parameter Prediction**:
> We clarify that the "parameter prediction" highlights the utility of the SonicMaster dataset metadata. Since the dataset contains the exact parameters used for generation, it can support future research into blind parameter estimation (e.g., predicting compressor settings). However, as this is not the primary objective of SonicMaster, we did not benchmark this specific task.
>
> ###### **Hidden Clipping**:
> We introduced "hidden clipping" to simulate real-world acoustic phenomena, such as constructive interference in reverberant environments or gain overcompensation after compression.
>
> ###### **Peak Normalization**:
> We normalize audio to the range \([-y, y]\) where y $\sim$ U(0.8, 0.9)\. Since degradations like compression and EQ significantly alter amplitude, peak normalization is essential to prevent unintentional clipping and standardize input levels. This serves as the control case opposite to "hidden clipping," where we intentionally omit normalization to force the model to learn declipping.
> Normalization is computed as:
>
> $x_{norm} = \frac{x_{degraded}}{\max(|x_{degraded}|)} \cdot y$
>
>
> We have updated the manuscript to incorporate changes and clarifications based on your feedback. Please have a look at the revised version. We hope our detailed responses and additional experiments address your concerns satisfactorily. We are happy to provide any further clarifications if needed. Thank you again for your valuable feedback.
>
> ###### **Papers worth mentioning**:
>
> We have added Rice et al. (2023) and Han et al. (2024) to the related works section as suggested.
>
> ###### **typos**:
> We have fixed all the typos pointed out by the reviewer.
>
> ###### **References**:
> [1] Rice, Matthew, et al. "General purpose audio effect removal."
>
> [2] Melechovsky, Jan, et al. "Mustango: Toward Controllable Text-to-Music Generation."

---

> ### Author Response · Authors · 2025-11-27
> **Follow-up on our review response**
>
> Dear Reviewer xvXv,
>
>
> As the discussion period progresses, we wanted to briefly follow up to ensure you had a chance to view our response.
>
> Based on your feedback, we have made substantial updates to the manuscript and our experimental setup. We believe our response above and the corresponding revisions directly address your primary concerns.
>
> We would value your feedback on whether they mitigate your concerns. We remain available for any further questions.
>
> Best regards,
>
> The Authors

---

### Official Review · Reviewer_Ej1n · 2025-10-30

**Soundness:** 4
**Presentation:** 3
**Contribution:** 4
**Rating:** 6
**Confidence:** 4

**Summary:**

The paper proposed a neural network called SonicMaster which is an unified generative model for music restoration and mastering. The proposed architecture is conditioned on text with instructions.
In order to train the model, SonicMaster, they build a dataset which contain paired degraded and high-quality audio examples by simulating them with nineteen degradation functions types.

The proposed architecture use flow-matching generative model that maps the input and the text prompts into the wanted audio file. The proposed model show improved sound quality across categories which also evaluate with human listeners.

**Strengths:**

1. The proposed SonicMaster is a single generative model that can handles a lots of artifact with guided text which is very practical solution

2. The dataset design is well organised, with 19 degradations type across five groups - EQ, dynamics, reverb, amplitude, stereo and also instruction alignment.

3. The paper used the well known Rectified Flow architecture with MM-DiT dual-stream that condition with FLAN-T5.

4. The results of the proposed architecture is very good compare to pervious works which can be seen in the multiple metrics - FAD, KL, SSIM and PQ

**Weaknesses:**

1. The human evaluation has very low amount of participants - only 13, more participants can strength the results

2. The results on EQ has the weakest performance among all categories which may be results of a problem of balancing

3. The proposed network trained only on synthetic data which may limit the results on real world data.

**Questions:**

1. Does the performance hold on real world scenario where the recording is more complex?

2. How much the proposed model sensitive to wording in the text conditioning?

3. What is the latency and the memory usage of the proposed model? Does the model size has trade off with the quality?

---

> ### Author Response · Authors · 2025-11-25
> **Response to Reviewer Ej1n (part 1/3)**
>
> Dear Reviewer,
>
> We would first like to thank you for providing us with insightful questions and feedback. We greatly appreciate the detailed feedback and insightful questions you have asked. In response, we have conducted significant additional human evaluation experiments. Kindly find our detailed response below:
>
> **Human Evaluation:** While eight participants may seem modest, it’s important to stress that evaluating subtle mastering degradations requires expert listeners with deep domain knowledge in music production. For context, our sample size already exceeds that of recent related work (e.g., [1] used only four experts under similar constraints). In response to your feedback, we recruited four additional experts during the rebuttal period, bringing the total to twelve participants. We have recalculated the results and updated Figure 4 in the revised manuscript to reflect the larger cohort, and the trends remain consistent.
>
> Additionally, note that we conducted a separate subjective evaluation of overall audio-quality comparing SonicMaster output with other baselines; while that test focuses more broadly on audio quality rather than mastering-specific degradations, it further supports our conclusions. This test was to this day attended by 20 participants and the results, which show a consistent trend to the results from the first version, are included in the revised version of the paper in Figure 5.
>
> We included Figure 4 to provide a clearer visual summary of the MOS results. It directly reflects the same 95%-CI values reported in the following table and allows easier comparison across degradation types. We can further refine the visualization in the camera-ready version if helpful.
>
> Table: Listening study - SonicMaster’s performance on specific degradations – MOS 95% CI
> | Category          | Text relevance      | Quality1            | Quality2            | Consistency         | Preference          |
> |-------------------|----------------------|----------------------|----------------------|----------------------|----------------------|
> | **EQ**            | 5.04 ± 0.37          | 4.48 ± 0.51          | 4.85 ± 0.38          | 4.83 ± 0.34          | 4.75 ± 0.32          |
> | **Reverb**        | 5.36 ± 0.52          | 4.18 ± 0.51          | 5.25 ± 0.33          | 5.18 ± 0.36          | 5.20 ± 0.47          |
> | **Dynamics**      | 4.44 ± 0.55          | 3.69 ± 0.58          | 4.94 ± 0.53          | 4.88 ± 0.55          | 4.94 ± 0.65          |
> | **Amplitude**     | 6.21 ± 0.41          | 3.42 ± 0.45          | 5.29 ± 0.48          | 5.40 ± 0.55          | 5.71 ± 0.52          |
> | **Stereo**        | 5.75 ± 0.75          | 4.79 ± 0.67          | 5.62 ± 0.45          | 5.46 ± 0.52          | 5.42 ± 0.57          |
> | **Mixed degradations** | 4.88 ± 0.73    | 3.58 ± 0.66          | 4.30 ± 0.58          | 4.25 ± 0.63          | 4.70 ± 0.52          |
>
> **EQ Performance Concern:** Thank you for highlighting that EQ shows the weakest performance among all categories. We agree with this observation in the listening study, where preference of EQ score is the worst among the categories. This might be due to two reasons: a) some of the effects are less audible and noticeable to the listener’s ear, e.g., airiness (defined as increase in spectral content above 10 kHz) or boominess (defined as increase in spectral content below 120 Hz); or b) the sole effect of “preference”, i.e., EQ mostly comprises stylistic change, which may not be to liking to each listener, unlike less ambiguous and clearer suppression of clipping or removal of reverb.
>
> However, we would like to point out that in objective evaluation the EQ category performs very well, showing clear signs of improvement over the degraded inputs as well as better performance when compared to baselines. Moreover, the second listening study shows clear preference of SonicMaster over the baselines of Text2FX and Mel2Mel+DiffWave in EQ.
>
> **Limitations of Synthetic-only Training:** We would like to highlight that our synthetic pipeline is built from physically grounded, real-world audio processes. Notably, we utilize real room impulse responses and microphone transfer functions during the degradation process, as well as many effects with a range of random parameters (e.g., X-band with 8 to 12 bins, each with a random dB gain/attenuation), which highly promotes robustness and generalizability by emulating real-world unpredictability (we stored these parameters in the dataset metadata for the purpose of reproducibility and other tasks, such as parameter prediction tasks).
>
> Additionally, we performed experiments beyond synthetic conditions: on real historical piano recordings, where no synthetic degradations were applied SonicMaster improves this audio across different metrics (Table 4, p. 9). This indicates that the model learns robust, degradation-invariant representations that transfer to real-world scenarios despite training only on controlled degradations.
>
> tbc in part2..

---

> ### Author Response · Authors · 2025-11-25
> **Response to Reviewer E1jn (part 2/3)**
>
> ...Continued from part 1...
>
> **Sensitivity to Prompt Wording:** Given the number of prompts used to train the model and the power of FLAN-T5, the model is robust to minor changes in the text conditioning. Given that the model works quite well without any text conditioning in its “auto” mode, as part of our ablation study (Tables 1, 2, and 3), a slight change in wording should in theory give results that are not worse than when using no text input.
>
> To further investigate the text controllability, we performed an additional ablation experiment. We used our test set and randomly shuffled the prompts in it. As expected, the performance is worse than when no prompts or the correct prompts were used, e.g., when targeting brightness, the properly text-guided model gave an average absolute error of 0.0059, no text prompt input resulted in an average absolute error of 0.0101, and shuffled prompts led to a further drop in performance with average absolute error of 0.0132. In KL, “correct” prompts resulted in KL of 0.696, no prompts gave KL of 0.917, and shuffled prompts gave KL of 2.014. The results of this experiment are now newly displayed in Tables 1, 2, and 3. The updated rows in corresponding tables are as follows for your reference:
>
> Table 1: EQ Objective Evaluation (Average Absolute Error).
> | **Model**                    | **Clarity** | **Boom** | **Airy** | **Bright** | **Dark** | **Muddy** | **Warm** | **Vocals** | **Mic.** | **X-band** |
> |------------------------------|-------------|----------|----------|------------|----------|-----------|----------|------------|----------|------------|
> | **SonicMaster (Ours)**       | **0.0114**  | **0.0834** | **0.0019** | **0.0059** | **0.0058** | **0.0388** | **0.0617** | **0.0576** | **0.0088** | **0.0358** |
> | Ablation — No Text Condition | 0.0130      | 0.1432   | 0.0032   | 0.0101     | 0.0086   | 0.0448    | 0.0841   | 0.0668     | 0.0154   | 0.0424     |
> | Ablation — Shuffled Prompts  | 0.0187      | 0.2075   | 0.0077   | 0.0132     | 0.0362   | 0.0981    | 0.1648   | 0.1043     | 0.0424   | 0.0998     |
>
>
> Table 2: Objective Scores: Reverb, Dynamics, Amplitude, and Stereo
>
> | **Model (MMDiT/DiT)**        | **Reverb – Small** | **Reverb – Big** | **Reverb – Mix** | **Reverb – Real** | **Dynamics – Comp.** | **Dynamics – Punch** | **Amplitude – Clip** | **Amplitude – Vol.** | **Stereo** |
> |------------------------------|--------------------|------------------|------------------|--------------------|------------------------|------------------------|------------------------|------------------------|-----------|
> | **SonicMaster (Ours)**       | **0.3663**         | **0.3726**       | **0.3935**       | **0.3109**         | **0.0193**             | **0.0871**             | **1.506**              | **0.0468**             | **0.1058** |
> | Ablation — No Text Condition | 0.3732             | 0.3805           | 0.4012           | 0.3264             | 0.0157                 | 0.0730                 | 2.812                  | 0.0465                 | 0.1416    |
> | Ablation — Shuffled Prompts  | 0.4161             | 0.4236           | 0.4538           | 0.3903             | 0.0225                 | 0.0895                 | 2.874                  | 0.0895                 | 0.3213    |
>
> Table 3: Objective Score  FAD (↓), KL (↓), SSIM (↑), and PQ (↑). KL
>
> | **Model**                    | **Single FAD ↓** | **Single KL ↓** | **Single SSIM ↑** | **Single PQ ↑** | **Double+Triple FAD ↓** | **Double+Triple KL ↓** | **Double+Triple SSIM ↑** | **Double+Triple PQ ↑** | **All FAD ↓** | **All KL ↓** | **All SSIM ↑** | **All PQ ↑** |
> |------------------------------|------------------|------------------|--------------------|------------------|---------------------------|---------------------------|----------------------------|---------------------------|----------------|----------------|------------------|----------------|
> | **SonicMaster (Ours)**       | **0.069**        | **0.696**        | **0.624**              | **7.743**        | **0.082**                 | **1.145**                 | 0.589                      | **7.654**                 | **0.073**      | **0.888**      | **0.609**            | **7.705**      |
> | Ablation — No Text Condition | 0.069            | 0.917            | 0.621              | 7.772            | 0.088                     | 1.484                     | 0.586                      | 7.643                     | 0.074          | 1.160          | 0.606            | 7.716          |
> | Ablation — Shuffled Prompts  | 0.081            | 2.014            | 0.598              | 7.610            | 0.131                     | 3.249                     | 0.558                      | 7.283                     | 0.098          | 2.543          | 0.581            | 7.470          |
>
> To be continued in part 3...

---

> ### Author Response · Authors · 2025-11-25
> **Response to reviewer E1jn (part 3/3)**
>
> ...Continued from part 2...
>
> **Latency, Memory Usage, Model Size & Quality Trade-off:** To contextualize the expected latency and memory profile, SonicMaster (≈550M parameters) generates 30 seconds of 44.1 kHz stereo audio in ~3.8 seconds on a single NVIDIA A40 GPU, and a full-length song typically takes about one minute.
>
> To clarify the relationship between model size and performance, we report model-scaling results from our ablation study (Tables 8–10, Appendix). We compare three capacities: Small (2 MM-DiT + 6 DiT blocks), Medium (4 MM-DiT + 12 DiT blocks), and Large (6 MM-DiT + 18 DiT blocks), and observe comparable or better performance with larger size. For instance, Boom stays similar for Large and Small sizes, but scored the best for a Medium size 0.0819 → 0.0698 → 0.0834 (Small → Medium → Large), but X-Band and Microphone see improved performance with size (0.0408 → 0.0383 → 0.0358 and 0.0122 → 0.0091 → 0.0088, respectively). The main difference can be seen in Clip, where the large model clearly outperforms the smaller sizes 2.363 → 2.455 → 1.506. Furthermore, the Large model shows the best performance in FAD, KL, SSIM and PQ across the board (Table 10). These results illustrate that larger models yield measurable quality gains, which naturally implies higher computational cost in both latency and memory consumption. For additional degradation types and detailed comparisons across model sizes, please refer to Tables 8–10 as part of Appendix A.7.
>
> We have updated the manuscript to reflect the aforementioned changes (subjective evaluation). Please have a look at the revised version. We kindly hope you find our responses address your concerns satisfactorily and will be happy to provide any further clarifications if needed. Thank you once again for your valuable feedback.
>
> [1] Melechovsky, Jan, et al. "Mustango: Toward Controllable Text-to-Music Generation."

---

> > ### Comment · Reviewer_Ej1n · 2025-11-26
> >
> > Thank you for addressing my concerns and for your valuable answer.

---

> > > ### Author Response · Authors · 2025-11-27
> > > **Thank you for your feedback**
> > >
> > > Dear Reviewer Ej1n,
> > >
> > >
> > > Thank you very much for your reply and for confirming that our response has satisfactorily addressed your concerns.
> > >
> > > We sincerely appreciate the time you took to review our work. Your feedback was instrumental in helping us strengthen our paper. We remain fully available to engage further should you have any additional questions or require more details as the discussion period continues.
> > >
> > > Best regards,
> > >
> > >
> > > The Authors

---

### Official Review · Reviewer_cmFV · 2025-11-01

**Soundness:** 4
**Presentation:** 4
**Contribution:** 4
**Rating:** 8
**Confidence:** 4

**Summary:**

This paper presents SonicMaster, a unified generative framework for controllable music restoration and mastering. The model addresses nineteen common degradation types (e.g., reverb, clipping, EQ imbalance) with a flow-matching architecture guided by natural-language prompts. It introduces a self-constructed text-conditioned dataset of degraded–clean audio pairs for training. Experiments show clear advantages over existing systems in both objective metrics and human listening tests.

**Strengths:**

The paper introduces the first unified model for prompt-based multi-artifact music restoration, addressing 19 types of degradations via flow-matching and classifier-free guidance. This unified formulation represents a significant step toward scalable, controllable, and generalizable audio restoration.

**Weaknesses:**

1.  Objective evaluation has some counter-intuitive metric behavior. For example, in Snippet single degradation in Table 3, SonicMaster's best FAD ($0.069$) and SSIM ($0.625$) scores are less favorable than the Degraded Input's FAD ($0.061$) and SSIM ($0.838$).

2. The model exhibits a clear performance drop in full-song inference compared with snippet in most metrics across Table 2 and Table 3.

3. Subjective evaluations (Table 5 and 6) do not include a quality rating for the Ground Truth (clean audio). The absolute quality rating in Table 5 does not include any SOTA baseline models for direct comparison.

**Questions:**

1. Would it be possible to include a subjective evaluation of the ground truth for comparison?

---

> ### Author Response · Authors · 2025-11-25
> **Response to Reviewer cmFV**
>
> Dear Reviewer,
>
> We would like to thank you for your feedback and positive assessment of this paper. We kindly respond to your points below:
>
> **Counter-intuitive FAD/SSIM behaviors:** Thank you for this observation. We would kindly like to re-emphasize that the audio quality, as reflected in the presented metrics, is negatively impacted by the VAE encoder-decoder processing. This degree of quality drop is reflected by the “reconstructed input” row in Table 3. As such, this row shows worse scores than that of the SonicMaster, indicating that the model itself improves the quality even through the VAE’s negative effect. Indeed, upgrading this VAE module could eliminate the negative effect of reconstruction, which is a matter of future work.
>
> **Full-song vs. snippet performance:** We acknowledge the performance reduction in long-form inference. This is primarily due to the current segment-linking strategy used to process 30-second windows. We are already exploring improved cross-segment conditioning to mitigate this issue in future work. Nonetheless, the full-song results still show substantial improvements over the degraded inputs across most metrics (Tables 1–3)
>
> **Subjective evaluation and absence of ground-truth ratings / baseline MOS:** Given that our model performs text-conditioned unified restoration and mastering, the listening test naturally focuses on text controllability and perceived improvement. The study is already demanding for expert listeners in both duration and required musical expertise. Adding MOS ratings for the clean ground-truth audio is less informative because these tracks are professionally mastered and consistently receive near-ceiling scores with minimal variance, offering no diagnostic value for comparing systems. Moreover, our evaluation targets relative enhancement of how well the model improves the degraded input according to the prompt rather than the absolute quality of the original music. Since the clean reference is not produced by any system, its MOS would not meaningfully contribute to the comparison. As such, we decided to continue the study with the current design.
>
> On that note, we managed to recruit 4 new participants for the first listening study (to current total of 12), and expanded the second listening study with an additional baseline and now report results with 20 participants (up from 13). These results are depicted in Figures 4 and 5 of the revised version.
>
> We included Figure 4 to provide a clearer visual summary of the MOS results. It directly reflects the same 95%-CI values reported in the following table and allows easier comparison across degradation types. We can further refine the visualization in the camera-ready version if helpful.
>
> Table: Listening study - SonicMaster’s performance on specific degradations – MOS 95% CI
> | Category          | Text relevance      | Quality1            | Quality2            | Consistency         | Preference          |
> |-------------------|----------------------|----------------------|----------------------|----------------------|----------------------|
> | **EQ**            | 5.04 ± 0.37          | 4.48 ± 0.51          | 4.85 ± 0.38          | 4.83 ± 0.34          | 4.75 ± 0.32          |
> | **Reverb**        | 5.36 ± 0.52          | 4.18 ± 0.51          | 5.25 ± 0.33          | 5.18 ± 0.36          | 5.20 ± 0.47          |
> | **Dynamics**      | 4.44 ± 0.55          | 3.69 ± 0.58          | 4.94 ± 0.53          | 4.88 ± 0.55          | 4.94 ± 0.65          |
> | **Amplitude**     | 6.21 ± 0.41          | 3.42 ± 0.45          | 5.29 ± 0.48          | 5.40 ± 0.55          | 5.71 ± 0.52          |
> | **Stereo**        | 5.75 ± 0.75          | 4.79 ± 0.67          | 5.62 ± 0.45          | 5.46 ± 0.52          | 5.42 ± 0.57          |
> | **Mixed degradations** | 4.88 ± 0.73    | 3.58 ± 0.66          | 4.30 ± 0.58          | 4.25 ± 0.63          | 4.70 ± 0.52          |
>
>
> The manuscript has been updated to reflect the aforementioned changes. Please have a look at it. We kindly hope you find our responses address your concerns satisfactorily and will be happy to provide any further clarifications if needed. Thank you once again for your valuable feedback.

---

> ### Author Response · Authors · 2025-11-27
> **Follow-up on our review response**
>
> Dear Reviewer cmFV,
>
> Thank you again for your positive assessment of our work. We also hope our explanations clarified your queries. We are happy to provide further details if needed.
>
> Best regards,
>
> The Authors

---

### Official Review · Reviewer_BVLP · 2025-11-04

**Soundness:** 2
**Presentation:** 4
**Contribution:** 2
**Rating:** 4
**Confidence:** 5

**Summary:**

The paper proposes a unified, text-guided generative system for controllable music restoration and mastering. The proposed model, SonicMaster, attempts to correct a broad set of music degradations in EQ, dynamics, reverb, amplitude, and stereo, using a rectified-flow model built with MM-DiT + DiT blocks operating in a VAE latent space. Evaluation includes degradation-aware metrics, global fidelity measures, and subjective listening tests, demonstrating that SonicMaster generally improves degradation inputs and is preferred by listeners against prior baselines.

**Strengths:**

The strengths of this paper lie in two parts.

1. Task Novelty. The paper draws a clear line between music restoration/mastering and adjacent areas (e.g., source separation, generation, enhancement), and argues for a single controllable model that handles multiple artifact types prompted by text instruction in an editing mode. This framing is insightful and latest for Audio AI, which has recently focused on diffusion/RFM generation and instruction-guided editing.

2. The analysis on the degradation taxonomy and the corresponding solution on simulation are well-specified. The dataset design is methodical: 19 degradations across five groups with explicit implementations, parameter ranges, and prompt templates. These definitions drive both training and evaluation, creating consistent interfaces between what’s degraded and what the model should fix. The model choice, conditional modules design, and experimental designs are well-organized. The rectified-flow-matching model is sensible for enhancement and editing in a generative manner. And the text conditioning via FLAN-T5 and audio pooling branch for segment stitching are practical, reproducible design decisions.

3. The experimental design is comprehensive. Ablation studies cover audio-text conditioning, sequence length, and model scale variants. Beyond global fidelity (FAD, KL, SSIM, PQ), the authors compute targeted artifact metrics and run two subjective listening test. This breadth reflects the ambition of authors to address the task in multiple angles.

**Weaknesses:**

However, the paper has three critical weaknesses that are worth discussing during the rebuttal process.

1. Despite the task novelty, the methodological novelty at the architecture/learning level is thin for a top-tier AI conference. Specifically, the model uses known blocks (a.k.a., VAE + MM-DiT/DiT + text encoder; rectified-flow objective; and classifier-free guidance), but does not propose new conditional interfaces (e.g., operator-aware adapters, band-mask tokens, time-varying schedules) nor new inference-time optimization strategies that specifically target enhancement failure modes in music context.

2. Despite the comprehensive experimental design, the comparisons to baselines miss close peers in instructional audio editing. Baselines emphasize classic signal-processing (WPE, HPSS) and a mel-spec enhancement pipeline (Mel2Mel+DiffWave) + Text2FX (EQ only). However, some advanced models such as AUDIT and MusicMagus, and more related instruction-guided models are only mentioned in passing without reporting results on implementations under the same data and prompts. This limits claims about data simulation effectiveness and model advantage over modern generative-based editing models. While category-specific metrics are valuable, several are proxy measures that can disagree with perception in dense mixes. Listening tests are helpful but small (8 for MOS; 13 for pairwise study). Generalization beyond the simulation (e.g., real recording faults, overall enhancement) remains under-explored to see if the model contains great generalization capabilities to understand "what is a good music quality" after learning millions of simulations in music degradation.

3. The demos in the website are not attractive and reflect some mis-leading in the optimization target.In the paper’s own analysis, reverb metrics on full-length songs are mixed, and SSIM/FAD sometimes decrease relative to inputs. But stereo and some EQ categories do not consistently improve. The observations in the demo website reinforce this: (1) de-reverberation sometimes adds tail energy rather than removing it (4th, 6th, 12th examples); (2) stereo edits can devolve into delays rather than proper image widening (9th, 13th examples); and (3) EQ correction can miss the target on more complex mixes (3st examples). (4) only simple operations such as leveling and bandwidth extensions perform relatively effective, while prior generative works can also lead the performance and this should also require a comparison to them. These gaps suggest simulation-to-real mismatch and insufficient inductive bias in the conditioning to constrain physically plausible edits.

4. Another minor comment on the data quality. Although the dataset is nominally sampled at 44.1 kHz, many training and evaluation tracks appear to have effective bandwidths only up to 32–36 kHz, showing striping artifacts that are typical of MP3 compression. This limits the model’s exposure to true full-band audio and its ability to restore or generate high-frequency details (e.g., brightness, air spot, and spatial shimmer). A controlled high-band test or uncompressed dataset evaluation is needed to confirm performance on the real full-band music.

**Questions:**

1. Can you include AUDIT (might be very similar) and MusicMagus, or any other recent audio editing models as baselines re-trained under your data and prompts? Even if limited to key operators (EQ/reverb/stereo), this would ground the comparison in modern generative-base editing models.

2. How sensitive is the system to wording? For example, if prompts underspecify targets (e.g., “reduce reverb”), how are some metrics like gain/RT60 change decided?

3. Can you evaluate on real studio/home recordings with annotated fixes to demonstrate transfer beyond synthetic degradations? And Also can you provide some results of out-of-domain prompts (from the vague one "make the music overall sounds better" to a specific non-trained one "remove the distortion")?

4. Is there any architecture-wise novelty you would like to emphasize compared to previous editing models? Or does the task definition novelty and the data simulation yield most contributions of the paper?

---

> ### Author Response · Authors · 2025-11-25
> **Response to Reviewer BVLP (part 1/4)**
>
> Dear Reviewer,
>
> We thank you for your very valuable and extensive feedback. Kindly find our responses below:
>
> **Methodological and Architecture Novelty:** We thank the reviewer for the thoughtful feedback. While SonicMaster builds on established generative components (VAE, MM-DiT/DiT, rectified flow), our work provides a methodological contribution by technically unifying multiple mastering tasks within one model.
>
> To the best of our knowledge, SonicMaster is the first unified model capable of handling 19 heterogeneous music degradations like EQ, dynamics, reverb, amplitude, and stereo within a single controllable rectified-flow framework. Achieving this unification required several non-trivial design choices:
> 1. a conditioning interface supporting multi-effect and composite degradation text instructions, for which we chose to use the instruction-tuned model of FLAN-T5, as previous studies showed its significant performance [1] in audio generation,
> 2. A pooled-audio conditioning branch enabling long-form restoration,
> 3. A new open data with pre and post mastering music  (19 effects, 5 classes, 175k data pairs) that permits joint learning without collapse.
>
> Importantly, our work provides unique empirical findings: consistent improvements across degradations (Tables 1–2), improved perceptual quality (Table 3), and strong generalization to out-of-domain historical piano recordings despite no domain-specific training (Table 4).
> Prior methods treat these degradations independently; SonicMaster demonstrates they can be addressed by a single model, enabling capabilities (e.g., composite degradation handling, zero-shot transfer) not possible with task-specific systems.
> Finally, although our work does not propose a new architectural block per se, it establishes a new problem formulation for controllable all-in-one music restoration and mastering and demonstrates a practical, scalable solution that outperforms specialized baselines (e.g., DPTNet, UMX, DCUNet, TCN, HDemucs) even on their targeted tasks (Fig. 3).
>
> **Additional Baselines:**
> **a) Effect Removal models:** We evaluated SonicMaster against the "General Purpose Audio Effect Removal" benchmark (including Demucs, TCN, DCUNet, DPTNet). As shown in the new Figure 3, SonicMaster achieves substantially higher SI-SDR scores (47.11 dB for Dynamics, 45.76 dB for Reverb) compared to the best baselines (approx. 20 dB). This demonstrates that our generative approach outperforms discriminative removal models on these tasks.
>
> #### **Comparison of SI-SDR (↑) for Dynamics and Reverb Removal**
>
> | **Approach**   | **Params** | **Dynamics (SI-SDR ↑)** | **Reverb (SI-SDR ↑)** |
> |----------------|-----------:|-------------------------:|-----------------------:|
> | *Baselines*    |            |                          |                        |
> | Input          | -          | 15.57                    | 9.30                   |
> | DPTNet         | 2.9M       | 16.95                    | 9.82                   |
> | UMX            | 6.3M       | 15.39                    | 11.39                  |
> | DCUNet         | 7.7M       | 13.80                    | 12.13                  |
> | TCN            | 10.0M      | 14.49                    | 13.25                  |
> | HDemucs        | 83.6M      | 20.08                    | 13.59                  |
> | *Proposed*     |            |                          |                        |
> | **SonicMaster**| **550M**   | **47.11**                | **45.76**              |
>
> b) Additionally, we include Text2FX-Directional as a prompt-driven audio editing baseline to assess how well models follow the same natural-language EQ instructions.
>
> ### EQ Objective Evaluation (Lower is Better)
>
> | **Model**            | **Clarity ↓** | **Boom ↓** | **Airy ↓** | **Bright ↓** | **Dark ↓** | **Muddy ↓** | **Warm ↓** | **Vocals ↓** | **Mic ↓** | **X-band ↓** |
> |----------------------|--------------:|-----------:|-----------:|-------------:|-----------:|------------:|-----------:|-------------:|----------:|-------------:|
> | **Degraded Input**   | 0.0238        | 0.3601     | 0.0049     | 0.0143       | 0.0893     | 0.4560      | 0.4345     | 0.2525       | 0.2393    | 0.1782       |
> | **Text2FX-Directional** | 0.0421     | 0.3977     | 0.0206     | 0.0143       | 0.3021     | 0.2602      | 0.5461     | 0.2517       | 0.6120    | 0.5038       |
> | **SonicMaster (Ours)** | **0.0114**  | **0.0834** | **0.0019** | **0.0059**   | **0.0058** | **0.0388**  | **0.0617** | **0.0576**   | **0.0088** | **0.0358**   |
>
> To be continued to part 2...

---

> ### Author Response · Authors · 2025-11-25
> **Response to Reviewer BVLP (part 2/4)**
>
> ...Continued from part 1...
>
> **Subjective Evaluation:** Given the feedback on the number of responses, we decided to continue gathering responses and have now reached 12 and 20 responses (up from 8 and 13) for the first and second listening study, respectively. We would like to note that the first listening study, given its focus on controllability in all the 19 aspects, requires listeners with good music understanding, experience, and familiarity with the musical terms used. As such, it is difficult to recruit a large cohort for this study.
> We included Figure 4 to provide a clearer visual summary of the MOS results. It directly reflects the same 95%-CI values reported in the following table and allows easier comparison across degradation types. We can further refine the visualization in the camera-ready version if helpful.
>
> Table: Listening study - SonicMaster’s performance on specific degradations – MOS 95% CI
> | Category          | Text relevance      | Quality1            | Quality2            | Consistency         | Preference          |
> |-------------------|----------------------|----------------------|----------------------|----------------------|----------------------|
> | **EQ**            | 5.04 ± 0.37          | 4.48 ± 0.51          | 4.85 ± 0.38          | 4.83 ± 0.34          | 4.75 ± 0.32          |
> | **Reverb**        | 5.36 ± 0.52          | 4.18 ± 0.51          | 5.25 ± 0.33          | 5.18 ± 0.36          | 5.20 ± 0.47          |
> | **Dynamics**      | 4.44 ± 0.55          | 3.69 ± 0.58          | 4.94 ± 0.53          | 4.88 ± 0.55          | 4.94 ± 0.65          |
> | **Amplitude**     | 6.21 ± 0.41          | 3.42 ± 0.45          | 5.29 ± 0.48          | 5.40 ± 0.55          | 5.71 ± 0.52          |
> | **Stereo**        | 5.75 ± 0.75          | 4.79 ± 0.67          | 5.62 ± 0.45          | 5.46 ± 0.52          | 5.42 ± 0.57          |
> | **Mixed degradations** | 4.88 ± 0.73    | 3.58 ± 0.66          | 4.30 ± 0.58          | 4.25 ± 0.63          | 4.70 ± 0.52          |
>
> **Demos:** Thank you for these observations. We note the samples can contain certain artifacts. While the FAD and SSIM metrics sometimes show “decrease” when compared to input, we would like to re-emphasize the negative effect of passing the audio through the VAE encoder-decoder pipeline, as reported by the “reconstructed input” row in Table 3, which shows substantially worse scores than the “degraded input” row. When comparing SonicMaster output performance to these inputs, the quality vastly increases. Indeed, this points to a future work direction of replacing/upgrading the VAE module to eliminate the negative effect of compression in the latent space.
>
> | **Model**              | **Single FAD ↓** | **Single KL ↓** | **Single SSIM ↑** | **Single PQ ↑** | **Double+Triple FAD ↓** | **Double+Triple KL ↓** | **Double+Triple SSIM ↑** | **Double+Triple PQ ↑** | **All FAD ↓** | **All KL ↓** | **All SSIM ↑** | **All PQ ↑** |
> |------------------------|------------------|------------------|--------------------|------------------|---------------------------|---------------------------|----------------------------|---------------------------|----------------|----------------|------------------|----------------|
> | Degraded Input         | **0.061**            | **3.859**            | **0.838**          | **7.321**            | **0.184**                     | **6.827**                     | **0.696**                  | **6.632**                     | **0.106**          | **5.131**          | **0.777**        | **7.026**          |
> | Reconstructed Input    | 0.139            | 3.990            | 0.574              | 7.172            | 0.290                     | 6.984                     | 0.507                      | 6.501                     | 0.196          | 5.273          | 0.546            | 6.885          |
>
> **Data Quality:** Thank you for this observation. SonicMaster used readily-available open music datasets. Given the licensing and accessibility many public music collections face, many of such collections provide their files as mp3 files (Free Music Archive, MTG-Jamendo). We utilized the data from the JamendoMaxCaps dataset [2], which is available only as mp3. However, while working with the data, we converted it back to a lossless compression format (FLAC) to prevent any further accumulation of compression artifacts when creating the degraded audio pairs from the ground truth samples. We recognize the reviewer’s concern and view it as a roadmap for improvement. In future work, we will actively seek out or collect higher-SR data. As a next step, we plan to augment our training set with datasets such as MUSDB18-HQ, MedleyDB, MAESTRO, etc.
>
> To be continued in part 3...

---

> ### Author Response · Authors · 2025-11-25
> **Response to Reviewer BVLP (part 3/4)**
>
> ...Continued from part 2...
>
> **Sensitivity to Prompt Wording:** We assume your question is regarding possible parameter specification on the input and the effect of prompt wording and prompts themselves on the incurred change. Our responses are written below. If you have any further questions, we will gladly respond.
>
> ***How much to change?:*** Since the model was trained with prompts as displayed in the Appendix A.8 (Table 11 in the revised version, the very last page), it currently does not support the specification of control parameters, as this was out of the current scope. Through the robust learning deployed, the model should have a sense of how much of a change to make without any text specification. This is further supported by the inclusion of the “auto” mode – the model was trained in a manner to allow inference with no prompts, in which it should automatically improve what it finds necessary.
>
> ***Controllability and sensitivity:*** Given the number of prompts used to train the model and the power of FLAN-T5, the model is robust to minor changes in the text conditioning. Furthermore, the model allows the use of no text input to make changes on its own as it finds fit. The results of this ablation study are depicted in Tables 1, 2, and 3.
>
> To further support the evaluation of text controllability, we included one new experiment in these tables. In this experiment, we used our test set and randomly shuffled the prompts in it. As expected, this experiment showed worse results in the attribute controllability metrics in Tables 1 and 2 compared to the no-prompt and correct-prompt experiments, e.g., in brightness, “correct prompts” 0.0059, no prompts give 0.0101, and the shuffled prompts give 0.0132 in the average absolute error metric. This difference can be seen across the majority of the metrics, e.g., KL 0.696 vs 0.917 vs 2.014, Reverb-real 0.3109 vs 0.3264 vs 0.3903, or X-band 0.0358 vs 0.0424 vs 0.0998. These results demonstrate that the instruction to the model does matter when it comes to specifying the correct task. However, minor word changes should not impact the choice of a task given the robust FLAN-T5 text representation and prompt variety used to train the model. The results of this experiment are now newly displayed in Tables 1, 2, and 3. The updated rows in corresponding tables are as follows for your reference:
>
> Table 1: EQ Objective Evaluation (Average Absolute Error).
>
>
> | **Model**                    | **Clarity** | **Boom** | **Airy** | **Bright** | **Dark** | **Muddy** | **Warm** | **Vocals** | **Mic.** | **X-band** |
> |------------------------------|-------------|----------|----------|------------|----------|-----------|----------|------------|----------|------------|
> | **SonicMaster (Ours)**       | **0.0114**  | **0.0834** | **0.0019** | **0.0059** | **0.0058** | **0.0388** | **0.0617** | **0.0576** | **0.0088** | **0.0358** |
> | Ablation — No Text Condition | 0.0130      | 0.1432   | 0.0032   | 0.0101     | 0.0086   | 0.0448    | 0.0841   | 0.0668     | 0.0154   | 0.0424     |
> | Ablation — Shuffled Prompts  | 0.0187      | 0.2075   | 0.0077   | 0.0132     | 0.0362   | 0.0981    | 0.1648   | 0.1043     | 0.0424   | 0.0998     |
>
>
> Table 2: Objective Scores: Reverb, Dynamics, Amplitude, and Stereo
>
> | **Model (MMDiT/DiT)**        | **Reverb – Small** | **Reverb – Big** | **Reverb – Mix** | **Reverb – Real** | **Dynamics – Comp.** | **Dynamics – Punch** | **Amplitude – Clip** | **Amplitude – Vol.** | **Stereo** |
> |------------------------------|--------------------|------------------|------------------|--------------------|------------------------|------------------------|------------------------|------------------------|-----------|
> | **SonicMaster (Ours)**       | **0.3663**         | **0.3726**       | **0.3935**       | **0.3109**         | **0.0193**             | **0.0871**             | **1.506**              | **0.0468**             | **0.1058** |
> | Ablation — No Text Condition | 0.3732             | 0.3805           | 0.4012           | 0.3264             | 0.0157                 | 0.0730                 | 2.812                  | 0.0465                 | 0.1416    |
> | Ablation — Shuffled Prompts  | 0.4161             | 0.4236           | 0.4538           | 0.3903             | 0.0225                 | 0.0895                 | 2.874                  | 0.0895                 | 0.3213    |
>
> To be continued in part 4...

---

> ### Author Response · Authors · 2025-11-25
> **Response to Reviewer BVLP (part 4/4)**
>
> ...Continued from part 3...
>
> Table 3: Objective Score  FAD (↓), KL (↓), SSIM (↑), and PQ (↑). KL
>
> | **Model**                    | **Single FAD ↓** | **Single KL ↓** | **Single SSIM ↑** | **Single PQ ↑** | **Double+Triple FAD ↓** | **Double+Triple KL ↓** | **Double+Triple SSIM ↑** | **Double+Triple PQ ↑** | **All FAD ↓** | **All KL ↓** | **All SSIM ↑** | **All PQ ↑** |
> |------------------------------|------------------|------------------|--------------------|------------------|---------------------------|---------------------------|----------------------------|---------------------------|----------------|----------------|------------------|----------------|
> | **SonicMaster (Ours)**       | **0.069**        | **0.696**        | 0.624              | **7.743**        | **0.082**                 | **1.145**                 | 0.589                      | **7.654**                 | **0.073**      | **0.888**      | 0.609            | **7.705**      |
> | Ablation — No Text Condition | 0.069            | 0.917            | 0.621              | 7.772            | 0.088                     | 1.484                     | 0.586                      | 7.643                     | 0.074          | 1.160          | 0.606            | 7.716          |
> | Ablation — Shuffled Prompts  | 0.081            | 2.014            | 0.598              | 7.610            | 0.131                     | 3.249                     | 0.558                      | 7.283                     | 0.098          | 2.543          | 0.581            | 7.470          |
>
>
> We have updated the manuscript with the aforementioned changes to address your feedback. Please have a look at the revised version. We kindly hope you find our responses address your concerns satisfactorily and will be happy to provide any further clarifications if needed. Thank you once again for your very thorough and valuable feedback.
>
>
> [1] Ghosal, Deepanway, et al. "Text-to-audio generation using instruction guided latent diffusion model." Proceedings of the 31st ACM International Conference on Multimedia. 2023.
>
> [2] Roy, Abhinaba, et al. "Jamendomaxcaps: A large scale music-caption dataset with imputed metadata." arXiv preprint arXiv:2502.07461 (2025).

---

> ### Author Response · Authors · 2025-11-27
> **Follow-up on our review response**
>
> Dear Reviewer BVLP,
>
> As the discussion period progresses, we wanted to briefly follow up to ensure you had a chance to view our response. Based on your constructive feedback regarding baselines and evaluation, we have added new baselines and expanded the subjective evaluation that reinforces the strength of SonicMaster.
>
> We believe these additions directly address your concerns. We remain available for any further questions.
>
> Best regards,
>
> The Authors

---

### Author Response · Authors · 2025-12-03
**Summary of Discussion Phase & Key Improvements**

Dear Area Chair,

We provide below a short summary of the discussion phase and key improvements made to the paper.

Across the board, reviewers highlighted these strong points of this paper:


**Novel task**: SonicMaster introduces a novel task of unified music restoration and mastering with text-controllability.

**Practical design**: The design of SonicMaster is practical and goes in the direction of human-interpretable control, which is a relevant trend in both research and creative workflows.


**Detailed and organised dataset**: The introduction of a new open-source dataset to enable this task was received well, reviewers highlighted the detail (variety in prompts, provided parameters for each sample) and organisation of this dataset.


**Thorough evaluation**: Evaluation of the performance is very thorough with many objective experiments including various ablation studies as well as two deployed listening studies.


We addressed all reviewer concerns, resulting in the following valuable improvements:

**Number of baselines**: Some reviewers questioned the number of baselines and their non-deep learning nature. We performed further experiments to compare our SonicMaster to more deep-learning based baselines, specifically audio effect removal models (DPTNet, UMX, DCUNet, TCN, and HDemucs), where SonicMaster demonstrated superior performance.

**Text2FX baseline**: Reviewer xvXv discussed Text2FX not being a fair comparison to SonicMaster, given its “non-instructive” nature. We further performed a comparison to new “Text2FX directional” model, which aligns better with the instruction following nature of SonicMaster. Results show SonicMaster follows EQ instructions more accurately.

**Listening study**: To address concerns regarding participant counts, we re-conducted both listening studies with a higher number of participants at the same time maintaining high standards for data quality, as evaluating music mastering/restoration requires specific domain expertise that limits the pool of qualified subjects compared to general audio tasks. Furthermore, for the second listening study (preference test over other baselines), we confirm statistical significance via a two-sided Binomial test for the Reverb condition ($p < 0.001$) and a Chi-Square Goodness of Fit test for the four-way EQ comparison ($\chi^2(3, N=200) = 472.9, p < 0.001$).

**Generalization to Real-World Data**: To address questions on training on synthetic data (Reviewers Ej1n, BVLP), we evaluated the model on out-of-domain historical piano recordings. Despite lacking domain-specific training, SonicMaster achieved performance comparable to specialized state-of-the-art baselines (BABE-2), demonstrating strong generalization.

**Wording sensitivity**: Reviewers asked about the system’s sensitivity to wording in the control prompt. To address their enquiry, we performed an additional ablation experiment in which we randomly shuffled the prompts in the test set and ran inference, alongside re-emphasizing the results of previous ablation study which used no control prompts. The results showed a clear decrease in performance compared to correct prompts, highlighting the system's precise steerability.

**Evaluation metrics**: One reviewer queried about the different metrics used for evaluation not being explained properly. In response, we clarified the description in the paper and included a whole detailed, and comprehensive Appendix section in which all the formulas and necessary parameters are given.

We hope you find this summary useful and wish you a non-stressful decision period. 🙂

---

### Meta-Review · Area_Chair_nNBv · 2026-01-02

**Summary:**

Two reviewers evaluated the paper as below the acceptance threshold (score 2, 4) and two evaluated it as above the threshold (scores 6, 8). While the reviewers generally appreciated the novelty of the proposed task, they raised several concerns, including: (1) limited technical novelty (BVLP); (2) lack of comparisons to several modern baselines and unfair comparison to other baselines (BVLP, xvXv); (3) underwhelming results on the demo page (BVLP); (4) limited evaluation on real (rather simulated) low-quality recordings (BVLP, Ej1n); (5) some results exhibiting lower quality than the degraded inputs (cmFV); (6) lack of evaluation of sensitivity to the text prompt (BVLP, Ej1n); (7) limited human evaluation (Ej1n, xvXv); (8) ill-defined task formulation (xvXv).

In their response, the authors clarified that the main contribution is the introduction of the task of all-in-one music mastering and the introduction of a new dataset. They further added comparisons to newer baselines, expanded the human evaluation, and clarified that the output quality is sometimes limited by the VAE and is not a fundamental limitation of the approach.

The AC views the paper as an interesting contribution. However, the AC agrees with the reviewers that the technical novelty is limited for a top-tier Machine Learning conference. The AC also agrees with some of the criticism about the quality of the results on the website, and in particular about the underwhelming results on the task of restoring old piano recordings (perceptually much worse than the dedicated BABE-2 model), which could indicate that the approach may not generalize well beyond simulated degradations.

**Reviewer Concerns:**

The rebuttal provided point-by-point answers to all the questions. The concerns of limited technical novelty and limited evaluation on real recordings still stand. In addition, two more minor concerns remain: (1) The authors claim that the method is not sensitive to variations in the text prompt, which describe the same instruction in different words, however they do not provide an experiment to support this claim. (2) The authors mention that the VAE sometimes sets an upper bound on the achievable quality, however this emphasizes the inherent limitation of the proposed approach which uses a latent flow model.

**Reviewer Scores:**

**BVLP: score 4.**

This is also the original score. Some of the reviewer’s concerns have been addressed but some remain, like technical novelty and generalization to non-simulated degradations.

**cmFV: score 8.**

This is also the original score.

**Ej1n: score 6.**

This is also the original score.

**xvXv: score 4**

The original score was 2. Some of the reviewer’s concerns were addressed. However, the AC believes that the reviewer would not be fully convinced by the author’s response regarding a possible partial overlap between the clean and degraded distributions.

---

### Decision · Program_Chairs · 2026-01-26

Reject